# Decellularized Porcine Conjunctiva in Treating Severe Symblepharon

**DOI:** 10.3390/jfb14060318

**Published:** 2023-06-08

**Authors:** Fengmei Shan, Xueying Feng, Jie Li, Sha Yang, Fuhua Wang, Weiyun Shi, Long Zhao, Qingjun Zhou

**Affiliations:** 1Eye Institute of Shandong First Medical University, Eye Hospital of Shandong First Medical University (Shandong Eye Hospital), State Key Laboratory Cultivation Base, Shandong Provincial Key Laboratory of Ophthalmology, School of Ophthalmology, Shandong First Medical University, Jinan 250012, China; elite1985@163.com (F.S.); xyfeng_m@126.com (X.F.); lijieliangmu@163.com (J.L.); yangs198902@163.com (S.Y.); 2Shandong Provincial Key Laboratory of Ophthalmology, State Key Laboratory Cultivation Base, Eye Institute of Shandong First Medical University, Qingdao 266071, China; hualinsixian@163.com (L.Z.); qjzhou2000@126.com (Q.Z.)

**Keywords:** decellularized extracellular matrix, symblepharon, oral mucosal transplantation, conjunctival reconstruction

## Abstract

This prospective study aimed to evaluate the effectiveness of decellularized porcine conjunctiva (DPC) in the management of severe symblepharon. Sixteen patients with severe symblepharon were enrolled in this study. After symblepharon lysis and Mitomycin C (MMC) application, tarsus defects were covered with residual autologous conjunctiva (AC), autologous oral mucosa (AOM), or DPC throughout the fornix, and DPC was used for all the exposed sclera. The outcomes were classified as complete success, partial success, or failure. Six symblepharon patients had chemical burns and ten had thermal burns. Tarsus defects were covered with DPC, AC, and AOM in two, three, and eleven cases, respectively. After an average follow-up of 20.0 ± 6 months, the anatomical outcomes observed were complete successes in twelve (three with AC+DPC, four with AC+AOM+DPC, and five with AOM+DPC) (75%) cases, partial successes in three (one with AOM+DPC and two with DPC+DPC) (18.75%) cases, and failure in one (with AOM+DPC) (6.25%) case. Before surgery, the depth of the narrowest part of the conjunctival sac was 0.59 ± 0.76 mm (range, 0–2 mm), tear fluid quantity (Schirmer II tests) was 12.5 ± 2.26 mm (range, 10–16 mm), and the distance of the eye rotation toward the opposite direction of the symblepharon was 3.75 ± 1.39 mm (range, 2–7 mm). The fornix depths increased to 7.53 ± 1.64 mm (range, 3–9 mm), eye movement was significantly improved, and the distance of eye movement reaching 6.56 ± 1.24 mm (range, 4–8 mm) 1 month after the operation; the postoperative Schirmer II test (12.06 ± 2.90 mm, range, 6–17 mm) was similar to that before surgery. Goblet cells were finally found in fifteen patients by conjunctival impression cytology in the transplantation area of DPC, except for one patient who failed. DPC could be considered an alternative for ocular surface reconstruction of severe symblepharon. Covering tarsal defects with autologous mucosa is necessary for extensive reconstruction of the ocular surface.

## 1. Introduction

Symblepharon is defined as an adhesion between the palpebral and bulbar conjunctiva. The etiology of symblepharon commonly includes diverse ocular surface diseases. The most prominent causes include chemical and thermal burns, as well as several autoimmune disorders, such as mucous membrane pemphigoid and Stevens–Johnson syndrome, etc. [1]. Complex conjunctival infectious diseases may also cause symblepharon [2]. The pathogenic effects of symblepharon are determined by its location and severity. In addition to its effect on ocular surface health through a number of pathogenic mechanisms, including a reduction in tear reservoir, interruption of tear flow and spread, blink-related microtrauma resulting from an irregular tarsal surface, and cicatricial entropion, it may cause inadequate blinking, lagophthalmos, and ocular motility restriction. Therefore, it can seriously damage the function and aesthetics of the eyes.

Despite the challenging and extremely complex treatment modalities, there is currently no standardized surgical treatment for symblepharon. In severe cases, symblepharon lysis invariably creates an extensive conjunctival defect that must be amended using a conjunctival substitute. Otherwise, re-adhesion of the bare exposed surfaces has a high potential. Several biological materials have been applied in previous therapies, including conjunctival grafts, amniotic membranes [3], oral mucosa [4], nasal mucosa [5], split-thickness skin grafts [6], and serial injections of 5-fluorouracil (5-FU) into the fornices [7]. Despite certain successes, several issues remain to be solved. Autologous conjunctiva (AC) is the best physiological choice for repairing conjunctival defects. However, obtaining sufficient autologous conjunctival flaps for treating the treatment of severe symblepharon is challenging. In addition, in autoimmune-mediated inflammatory conditions, such as ocular cicatricial pemphigoid or Stevens–Johnson syndrome, any trauma to the conjunctiva can reactivate the underlying inflammatory process and should, therefore, be avoided. Amniotic membrane (AM) is the most widely used biological substrate for conjunctival reconstruction because of its inherent ability to promote epithelialization [8,9]. However, the availability, cost, and standardization of AM preparations remain challenging [10,11]. AM degrades quickly in an inflammatory environment, leading to a decreased chance of epithelialization [12]^.^ Its therapeutic effect is unstable during long-term observation, especially in cases of large-scale conjunctival defects [13]^.^ Grafts of the oral mucous membranes are widely used for fornix reconstruction [14]. Oral mucosa is easily available, and complications are uncommon; however, there are cosmetically apparent differences in the bulk, tint, and quality of the tissue with bulbar conjunctiva. When applied to a tarsus defect, the oral mucosa can maximize its function and overcome the aforementioned limitations. In addition, although autologous nasal was successfully applied in the reconstruction of the fornix [15], harvesting the nasal mucosa remains a complicated and challenging task [16].

The growing field of tissue engineering offers promising alternatives for overcoming these challenges. Decellularization of xenogeneic tissues or organs is a promising yet challenging biological engineering scaffold for transplantation [17]. We recently reported the use of decellularized porcine conjunctiva (DPC) for conjunctival reconstruction in rabbit models and in small clinical cases, which demonstrated enhanced transplant stability and improved epithelial regeneration in severe ocular surface damage compared with AM [18]. Due to the short-term postoperative follow-up and the small sample size of clinical cases, its efficacy was not fully evaluated. We further expanded the clinical application of DPC for the management of eyes with severe symblepharon and obtained preliminary clinical results.

## 2. Materials and Methods

### 2.1. Ethics

All clinical applications performed in this study were approved by the Ethics Committee of the Shandong Eye Hospital. All patients signed an informed consent form in accordance with the tenets of the Declaration of Helsinki for research involving human subjects.

### 2.2. Patients

Sixteen patients (16 eyes) with severe symblepharon were included in this non-comparative case series, and underwent a combined procedure with DPC in our hospital between August 2018 and January 2022. All the patients were monitored through follow-up for >6 months. The severity of symblepharon was grade III or higher, according to the grading criteria reported by Kheirkhah [19]. Surgery was performed at least six months after the chemical/thermal burns to allow for stabilization of ocular surface inflammation. Those who underwent previous symblepharon lysis surgery were observed for at least 6 months. Patients with systemic immune diseases were treated until the systemic condition was stable and no active eye inflammation was observed for at least six months before surgical treatment. Patients with severe dry eye despite previous punctual occlusion were excluded from this study. Other alternatives, including the possible advantages and shortcomings of the material, were thoroughly explained to the patients before surgery.

### 2.3. Preparation and Assessment of the Biomaterial

Duroc swine without a viral infection or medical history were selected. The whole conjunctivas were aseptically isolated from porcine eyes aseptically within 1–3 h postmortem. The epithelium of native conjunctiva was carefully scraped by epithelial scraper, and the tissue pieces were cleaned and incubated in super nuclease (400 U/mL, Sino 7 Biological Inc., Beijing, China) and 1% TrionX-100 at 37 °C for 2 h. After repeated washing, the prepared DPCs were sterilized using γ-irradiation (8 kGy; Zhongjin Irradiation, Qingdao, China) (Figure 1A). All the DPCs were prepared in different sizes and stored at −20 °C until use [18]. The animal experimental procedures were approved by the Ethics Committee of the Shandong Eye Institute, and all DPCS in this study were supplied by the Shandong Eye Institute. The DPCs should be assessed using H&E staining, Scanning electron microscope and collagen content prior to storage (Figure 1B–G). 

### 2.4. Surgical Techniques

All operations were performed by one surgeon. All adhesions and pseudopterygium were meticulously detached from the corneal limbus to expose the sclera. Extensive resection of the subconjunctival fibrous vascular tissue and scar tissue was performed to ensure free eye movement. After cautery of bleeding vessels slightly, several small surgical sponges that soaked in 0.04% MMC were inserted into the deep fornix for 5 min, followed by blotting of the excess liquid to ensure that the sponges did not come contact the sclera and cornea. Subsequently, care should be taken to ensure that no MMC sponges remained inside the deep fornix and were immediately removed. Finally, the fornix was rinsed adequately with 200 mL balanced salt solution [19].

For eyes with adequate residual conjunctiva to cover the palpebral conjunctival defect (grade IIIa), the cicatricial and fibrovascular tissues under the remaining conjunctiva and pseudopterygium were removed and fixed inside the surface of the tarsal plate (Figure 2A) [20]. In eyes with less conjunctiva remaining (grade IIIb and IIIc), the residual conjunctival was used to cover the palpebral conjunctival defect near the margin and the rest area would be mended with autologous oral mucosa (AOM) (Figure 2B). In eyes with atresia conjunctiva (grade IV), the palpebral conjunctiva defect was covered with AOM (Figure 2C). All the grafts in the deep fonix were fixed with one or two transcutaneous double-armed 1–0 silk threads.

AOM was taken from the lower lip as thinly as possible, 30% larger than the tarsal conjunctival defect, and soaked in 1:1000 gentamycin sulfate injection (10 mL strokephysiological saline solution with 10,000 u gentamycin sulfate injection) for 10 min. Subsequently, the oral mucosa was used to cover the tarsus with one side sutured to the residual conjunctiva or the lid margin using 10–0 nylon sutures. The other side was fixed deeply into the fornix with transcutaneous sutures (Figure 2B,C). 

DPC was trimmed after rehydration according to the size and shape of the bulbar and fornix conjunctiva defect,,and then transplanted onto the defect with the epithelium side upward. 10–0 nylon sutures were used to fix the DPC firmly to the superficial sclera interruptedly., The lower margin of the DPC was sutured end to end with the free end of residual conjunctiva (Figure 2A) or AOM (Figure 2B,C) on the conjunctiva. 

In patients with pseudopterygium, the corneal defect after removing the pseudopterygium was covered with AM.

### 2.5. Postoperative Treatment

The 10–0 nylon sutures were removed one week after surgery, while the 1–0 silk thread was removed another week later. All patients received gatifloxacin eye drops (qid) and recombinant bovine basic fibroblast growth factor eye gel (tid) postoperatively, and ofloxacin eye ointment was used once every night until inflammation resolved. In eyes with obvious conjunctival inflammation, preservative-free artificial tears and topical glucocorticoids were administered 1 week after surgery. The oral incision was healed by gargling with an antibacterial mouthwash thrice a day. All patients were revisited at 1 week, 2 weeks, 1 month, 3 months, and 6 months after surgery, as needed. 

### 2.6. Evaluation of Clinical Outcomes

Preoperatively, all patients underwent slit-lamp examination, Schirmer II test, fornix depth measurement, and ocular motility. Postoperatively, epithelialization of DPC was assessed using fluorescein staining. The Schirmer II test, fornix depth measurement, and ocular motility test were performed 1 month postoperatively. Conjunctival impression cytology was performed at the center of the DPC graft to evaluate goblet cells 1 month after surgery. All cytological specimens were collected by a single doctor, and evaluated and diagnosed by a single pathologist. According to Nelson’s method [21], all the specimens were evaluated using an optical microscope. 

#### 2.6.1. Fornix Depth Measurement

Proxymetacaine hydrochloride 0.5% was instilled once into the eyes one minute in advance. A flexible plastic ruler with a zero–tip scale was placed into the bottom of the fornix at the narrowest part of the conjunctival sac. Then the patients were instructed to turn his/her eye to the opposite direction in which the fornix measuring device was located. Depth measurements were obtained by identifying which marks aligned with the posterior lid margin [22]. A month after surgery, the same examiner used the same method to measure the fornix depth at the same location.

#### 2.6.2. Schirmer II Test 

Preoperative and 1–month postoperative Schirmer II tests were performed using the conventional method to assess the quantity of tear fluid. Patients with lower conjunctival sac atresia before surgery were not carried out.

#### 2.6.3. Measurement of the Amplitude of Eye Movement 

The patient was instructed to hold their head against the chinrest and forehead support on the slit–lamp, and a scale was used to mark the pupillary center, with the patient looking at the frontal visual target. The central pupil activity distance (mm) was measured by instructing patients to look in the direction opposite of the symblepharon. A patient whose pupil was covered with a pseudopterygium was able to record the movement distance of the pseudopterygium. The procedure was repeated thrice by the same examiner before and one month after surgery, and the average value was recorded.

The outcome was defined as complete success (restoration of an anatomically deep fornix without scarring or motility restriction), partial success (focal recurrence of scarring), or failure (return of the symblepharon) [19].

## 3. Results

This study included 16 eyes of 16 patients (15 men and 1 woman) with a mean age of 32.1 ± 13.6 years (range, 7–62 years). The etiology of symblepharon comprised thermal (n = 10) and chemical burns (n = 6). Eleven patients had undergone one to three surgical procedures for symblepharon lysing. The severity of symblepharon based on length was grade III in seven eyes (43.75%) and grade IV in nine eyes (56.25%). The symblepharon width was graded A in five eyes (31.25%), B in six eyes (37.5%), and C in five eyes (31.25%). The severity of conjunctival inflammation was 1+ in three eyes with grade III and three eyes with grade IV, 2+ in three eyes with grade III and six eyes with grade IV, and 3+ in one eye with grade III.

Surgery was performed uneventfully in all cases. The sclera was covered with DPC in all eyes. Tarsus defects were treated with AOM alone in seven patients (43.75%), AC alone in three patients (18.75%), AC and AOM in four patients (25%). As two patients (12.5%) refused oral mucosa excision, DPC was used in mending their tarsus defects.

The average postoperative follow-up was 20.0 ± 6 months (range, 10–32 months). The depth of the narrowest part of the conjunctival sac was 0.59 ± 0.76 mm (range, 0–2 mm) before surgery, and had a visible increase of 7.53 ± 1.64 mm (range, 3–9 mm) at the last follow up. Eye movement was significantly improved from 3.75 ± 1.39 mm (range, 2–7 mm) to 6.56 ± 1.24 mm (range, 4–8 mm) 1 month after operation. Schirmer II test is 12.06 ± 2.90 mm (range, 6–17 mm) one month after operation, which was similar to that before surgery (12.5 ± 2.26 mm, range 10–16 mm). 

In total, the anatomical outcomes included complete success in 12 eyes (75%), partial success in 3 eyes (18.75%), and failure in 1 eye (6.25%). In eyes with grade III symblepharon (n = 7), complete success was achieved in seven eyes (100%) and none developed failure (Figure 3A,B and Figure 4A,B). In eyes with grade IV symblepharon (n = 9), these outcomes were observed in five eyes with complete success (55.56%) (Figure 3C–F), three eyes with partial success (33.33%) (Figure 3G,H), and one eye with failure (11.11%). Table 1 provides a summary of the anatomical results for different types of symblepharon, and the details of all patients are shown in Table 2.

Conjunctival epithelialization of DPC occurred 1–2 weeks after operation (Figure 4B,C) and conjunctival impression cytology revealed goblet cells at the centre of the DPC graft one month after operation (Figure 4D) in 15 patients except one (case 10) who was stubbornly addicted to his mobile phone for more than 10 h every day and did not follow the doctor’s advice for regular medication and outpatient review. When he visited the outpatient clinic for a follow-up visit 4 weeks after operation, recurrence of symblepharon had been occurred due to the complete dissolution of the transplanted DPC and the development of cicatricial entropion.

## 4. Discussion

Conjunctival regeneration is a vital component of ocular surface reconstruction, particularly in patients with extensive conjunctiva involvement after severe symblepharon lysis. An ideal conjunctival substitute should meet several criteria, including a flexible matrix with good long-term elasticity, stability and tolerance, an epithelial layer with self-renewal potential on the surface of the matrix, and an epithelium that contains both epithelial and goblet cells [16]_._ In addition, it is important for substitutes to be easily accessible. Patients with severe burns often experience binocular damage and cannot provide autologous conjunctiva, while the number of donated human conjunctiva is limited. The extracellular matrix protein of pigs is homologous to that of humans and porcine conjunctiva is extremely easy to obtain; thus, pigs are increasingly being used as the source of allogeneic tissues. We previously took up an innovative approach to prepare the DPCs. The intricate conjunctiva-specific structures and abundant matrix components were preserved in the DPC, which offered favorable mechanical properties for the graft. DPC was shown to positively affect ocular surface repair, particularly in a rabbit model of severe symblepharon. The conjunctiva reconstructed using DPC exhibited epithelial heterogeneity, resembling that of the native conjunctiva. In addition, results from clinical studies were encouraging for pterygium and symblepharon, and the clinical application of DPC is promising [18]. In this study, DPC was clinically used to treat symblepharon, and 16 patients with severe symblepharon were included. The success rate was 75%, which is similar to that reported previously [4]. The severity of symblepharon in all 16 patients was grade III or IV, and the lesions occupied more than one quadrant of the conjunctival sac, suggesting that DPC could be an effective conjunctival substitute for the treatment of severe symblepharon.

Ocular reconstruction for severe symblepharon provided more than a good cosmetic appearance with a deep fornix but the formation of normal cell and tissue types, including goblet cells, and prevention of postoperative scar formation were equally important. Reconstruction with the AM can be successful; however, it tended to shrink when there was a persistent inflammation occurred after transplantation. Solomon et al. reported fornix reconstruction with AM in 12 of 17 patients and found that the best results occurred in eyes with symblepharon following trauma. Conversely, fornix contractions tended to recur due to an active inflammatory autoimmune disorder [23]. Previous animal studies found that, compared to AM, DPC had better extensibility, elasticity, and stability [24]. Stratified epithelium was observed in the eyes grafted with DPC 10 days after surgery; in contrast, the epithelialization of the AM transplanted area was manifested by contraction of the wound margin and growth of scar tissue instead of really epithelialization [24]. In addition, blood-filled vessels were visible in the early post-transplantation phase in DPC and were not observed in AM. Vascularization is crucial for the survival of transplants over long periods [25].

Witt found that DPC can promote goblet cell and epithelial cell regeneration [25]. In this study, impression cytology was performed to further confirm the epithelialization of conjunctival defects. A small number of goblet cells interspersed among numerous conjunctival epitheliums were detected. All these may benefit from the conjunctival matrix reserved in the DPC, which is conducive to the growth of cells. Considering the fragility of the new conjunctival epithelium, impression cytology of the ocular surface was not performed with caution until 4 weeks after the surgery to minimize the adverse effects of invasive operations. The authors reasonably believe that goblet cells may grow into DPC much earlier at a higher density. 

Various surgical techniques, including anchoring sutures, were developed to treat symblepharon [23]. In particular, anchoring sutures can prevent graft contracture and improve the success rate. Combined approaches of MMC, anchoring sutures, AM, and (or) AOM were designed according to the severity of symblepharon, and the results were satisfactory [26]. In addition, the analysis revealed that complete success was significantly positively correlated with intraoperative MMC use [19]. Considering the severity of symblepharon in the patients in our study, a combined approach with intraoperative MMC was selected. However, MMC is associated with complications such as scleritis, keratitis, and scleral melting [27]. In this study, MMC was placed in the subconjunctival fibrous tissue of fornix and not the sclera and was washed thoroughly with water. No relevant complications were found in all patients during the follow-up. However, determining the optimal concentration of MMC requires a long-term clinical research. Additionally, 5-FU was extensively used in glaucoma filtration surgery for years to inhibit fibroblast proliferation [28]. Jovanovic used 5-FU to reduce conjunctival scarring caused by systemic diseases, and achieved ideal results [7]; therefore, subconjunctival injection of 5–FU may be a new option for preventing the recurrence of symblepharon with fewer complications.

Kheirkhah reported amniotic membrane transplantation (AMT) alone to reconstruct the fornix with a high percentage of success in eyes with grades I and II symblepharon, but limited success in eyes with grades III and IV symblepharon [19]. He concluded that in severe symblepharon, additional mucosal grafting is needed to improve the curative effect and that, in addition to the autologous conjunctiva, AOM may be an appropriate option. In this study, DPC combined with AOM or AC achieved complete success in patients with grade III symblepharon. However, two patients refused autologous oral mucosa transplantation, and the tarsus defects were amended with DPC. This resulted in partial success with local recurrence of symblepharon, probably due to the large area of DPC transplantation, inability to epithelialize in a short time, and gradual autolysis of DPC, leading to adhesion recurrence. This further confirms that DPC combined with AOM or AC can effectively treat severe symblepharon. In this study, large–scale DPC transplantations were only partially successful, possibly because the grafts took prolonged periods to undergo complete epithelialization. For this reason, it is not recommended to use DPC to replace both the bulbar conjunctiva and tarsal defects in severe grade IV symblepharon.

Unexpectedly, we found a patient with mobile phone addiction who used his smartphone for >10 h every day and even forgot to take medications as prescribed. The transplanted DPC dissolved 4 weeks after surgery, and partial recurrence of symblepharon with cicatricial entropion eventually occurred. Hence, we believe that a moist ocular surface environment is crucial for the survival of conjunctival grafts and that patients should be educated to restrict their smartphone use.

In conclusion, DPC could be a suitable material for conjunctival reconstruction of severe symblepharon. The ideal conjunctival sac and ocular surface environment can lay the foundation for vision-improving surgery, such as lamellar keratoplasty, penetrating keratoplasty, or keratoprosthesis [29], Despite the expected success of DPC in clinical use, no randomized controlled studies were conducted, and objective measures of the physiological performance of DPC after transplantation are lacking. In future studies, we will examine the long-term effects of DPC and evaluate its physiological effects.

## Figures and Tables

**Figure 1 jfb-14-00318-f001:**
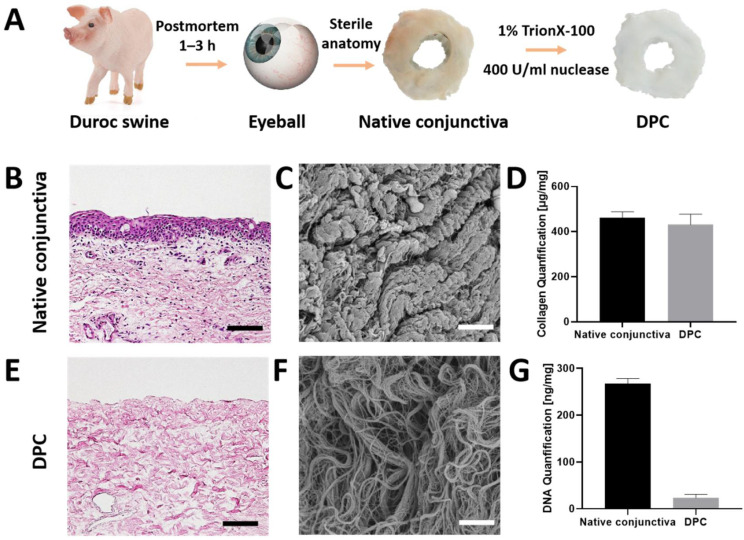
Duroc swine without virus infection and medical history were selected. The conjunctivas were isolated from porcine eyes aseptically within 1–3 h postmortem. The native conjunctivas without epithelium were incubated in super nuclease with 400 U/mL and 1% TrionX-100 at 37 °C for 2 h. After repeated washing, the prepared DPCs were sterilized and stored at −20 °C (**A**). Histological comparison of native porcine conjunctiva (NPC) (**B**) and decellularized conjunctiva (**E**) by H&E staining. Scale bar, 50 μm. Scanning electron microscope image of NPC (**C**) and DPC (**F**), scale bar, 10 μm. Comparison of collagen content between NPC and DPC (**D**), comparison of DNA content between NPC and DPC (**G**).

**Figure 2 jfb-14-00318-f002:**
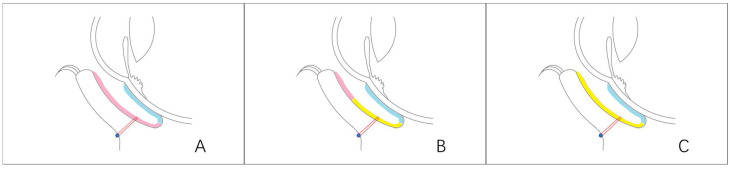
Schematic graphic of three different kinds of surgical strategies during reconstruction of conjunctival fornix. The conjunctival defect was mended with DPC (blue) from the corneal limbus to the deep fornix.The palpebral conjunctival defect was mended with residual conjunctiva (pink) (**A**), residual conjunctiva and AOM (yellow) (**B**) or totally AOM (**C**). Transcutaneous double-armed 1–0 silk threads were used to fixed graft deep into the fornix (red).

**Figure 3 jfb-14-00318-f003:**
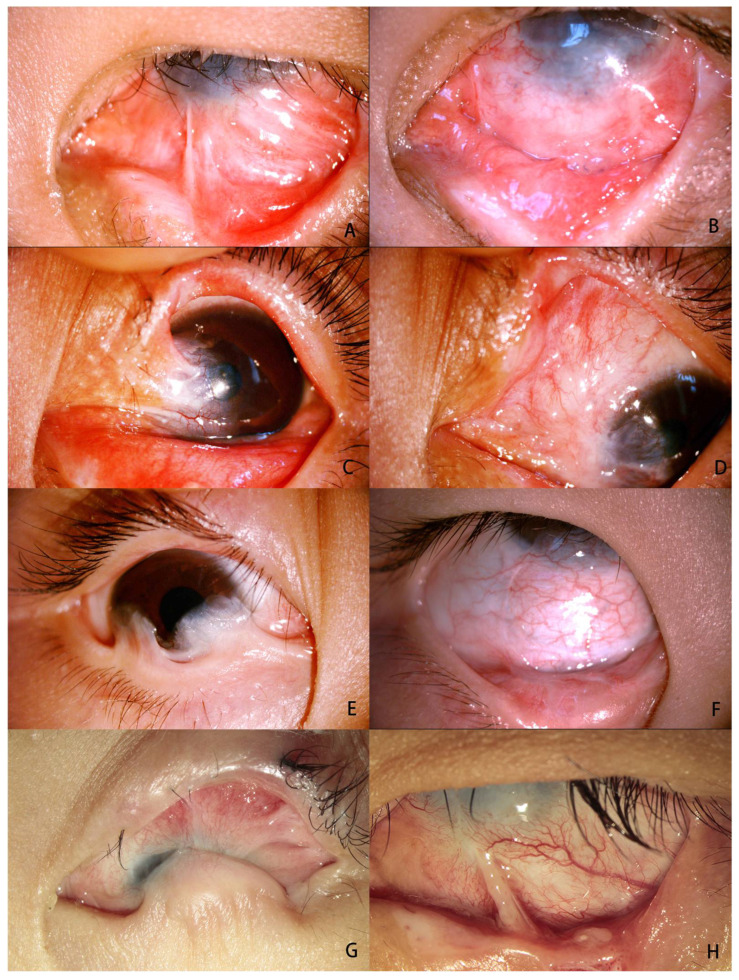
Photos showing severe symblepharon before and after fonix reconstruction with DPC. Case 7 with grade Ⅲ symblepharon in the inferior fornix (**A**) and complete success achieved with deep fornix one month after operation (**B**). Case 9 with grade IV symblepharon in the supranasal fornix (**C**) and complete success achieved with deep fornix 12 months after operaton (**D**). Case 12 with grade IV symblepharon in the inferior fornix with pseudopterygium and upgaze restriction (**E**), and a deep fornix without inflammation and upgaze restriction was observed at 11-month after operation (**F**). Case 2 with extensive grade IV symblepharon in the inferior fornix (**G**), and partial success was achieved with focal recurrence of symblepharon 6-month after operation (**H**).

**Figure 4 jfb-14-00318-f004:**
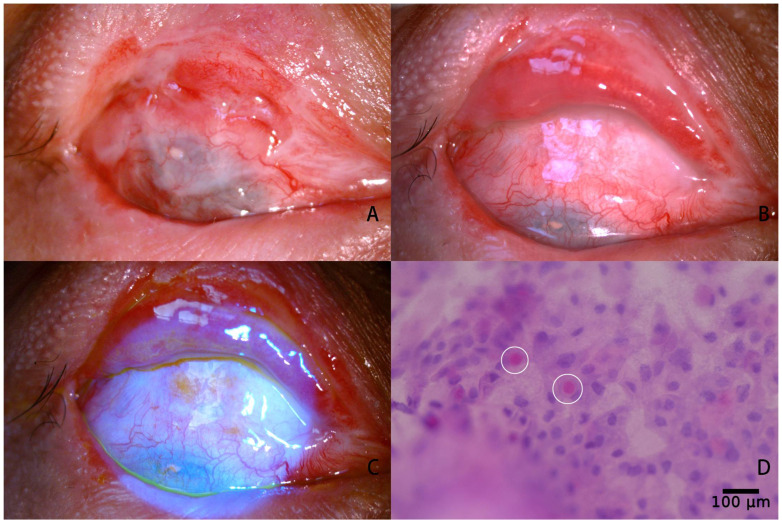
Photos of case 6 before and after fornix reconstruction. The upper conjunctival sac exhibited third-degree C symblepharon, accompanied by slightly hyperemic, pseudopterygium and restricted downward gaze (**A**). One month after operation, the conjunctival sac was significantly deepened and the residual autologous conjunctiva, AOM, and DPC all survived. The hyperemia of the ocular surface was reduced and down gaze restriction was released (**B**). Fluorescein staining shows complete epithelization of DPC graft (**C**). Goblet cells were found using impression cytology 1 month after surgery (the white circle) (**D**).

**Table 1 jfb-14-00318-t001:** Anatomical Outcomes of Fornix Reconstruction Using Residual Autologous Conjunctiva (AC), Autologous Oral Mucosa (AOM) and Decellularized Porcine Conjunctiva (DPC) Transplantation According to the Grade of Symblepharon.

Type of Symblepharon	Conjunctival Substitute	Number of Eyes	Clinical Outcomes			The Average of FD (mm)		The Average of Schirmer II Test (mm)		The Average of EMD (mm)		Goblet Cells (n)
			Complete Success, n (%)	Partial Success, n (%)	Failure, n (%)	Preoperative	Postoperative	Preoperative	Postoperative	Preoperative	Postoperative	
grade III	AC+DPC	3	7 (100)	0 (0)	0 (0)	1.36	8	12.5	14	4.86	6.93	7
	AC+AOM+DPC	4										
grade IV	DPC+DPC	2	5 (55.56)	3 (33.33)	1 (11.11)	0	7.17	none	10.56	2.89	6.28	8
	AOM+DPC	7										
Total		16	12 (75)	3 (18.75)	1 (6.25)	0.59	7.53	12.5	12.06	3.75	6.56	15

FD: Fornix depth. EMD: eye movement distance. Goblet cells (n): number of patients found to have goblet cells per group. mm: millimeter.

**Table 2 jfb-14-00318-t002:** Patients’ Data.

Number	Trauma Causes	Grading and Width of Symblepharon	Conjunctival Substitute (Palpebral + Bulbar Conjunctiva)	Clinical Outcomes	Preoperative FD (mm)	Postoperative FD (mm)	Preoperative Schirmer II Test (mm)	Postoperative Schirmer II Test (mm)	Preoperative EMD (mm)	Postoperative EMD (mm)	Goblet Cells
1	Alkali burn	IVb	DPC+DPC	partial success	0	8	Not examined	11	3	6.5	exist
2	thermal burn	IVb	DPC+DPC	partial success	0	9	Not examined	10	3	5	exist
3	thermal burn	IVa	AOM+DPC	complete success	0	7.5	Not examined	8	3.5	6	exist
4	Alkali burn	IIIc	AC+AOM+DPC	complete success	2	8	10	13	4	6	exist
5	Alkali burn	IIIa	AC+DPC	complete success	2	8	16	15	5	6.5	exist
6	thermal burn	IIIc	AC+AOM+DPC	complete success	1.5	8	13	15	3	5.5	exist
7	thermal burn	IIIa	AC+DPC	complete success	1	9	11	13	7	8	exist
8	thermal burn	IIIc	AC+AOM+DPC	complete success	1	6	Not examined	12	6	7	exist
9	thermal burn	IVa	AOM+DPC	complete success	0	5	Not examined	13	3.5	8	exist
10	Alkali burn	IVc	AOM+DPC	failure	0	3	Not examined	6	2.5	4	none
11	thermal burn	IVb	AOM+DPC	complete success	0	9	Not examined	10	2	7	exist
12	thermal burn	IVb	AOM+DPC	complete success	0	7.5	Not examined	13	3	7	exist
13	thermal burn	IVb	AOM+DPC	complete success	0	8.5	Not examined	15	3.5	8	exist
14	Alkali burn	IIIa	AC+DPC	complete success	1	9	14	17	4	7.5	exist
15	Alkali burn	IVc	AOM+DPC	partial success	0	7	Not examined	9	2	5	exist
16	thermal burn	IIIb	AC+AOM+DPC	complete success	1	8	11	13	5	8	exist

AC: residual autologous conjunctiva; AOM: autologous oral mucosa; DPC: decellularized porcine conjunctiva; FD: fornix depth; EMD: eye movement distance; mm: millimeter.

## Data Availability

Not applicable.

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
