# Peer review of "Decellularized Porcine Conjunctiva in Treating Severe Symblepharon"

_jfb, 2023, doi:10.3390/jfb14060318_

Round 1

Reviewer 1 Report

This is a case series of treatment of conjunctival scarring (symblepharon) in 16 eyes using decellularized porcine conjunctiva (DPC). 

Introduction:  Consider adding comment on the use of 5-fluorouracil injection for treatment of symblepharon (Jovanovich. Ophthalmic Plas Reconst Surg 2021;37:145). 

Methods: Describe more fully the harvesting of the porcine conjunctiva.  What is the size (area) of the tissue removed.  How was the epithelium removed?  Describe also the size of the DPC grafted tissue pieces?  Give a reference for the use of the bovine fibroblast growth factor post-op.

Results:  Include the mean area of DPC applied.  The outcome of "deep fornix" as a measure of complete success should be better defined. The use of keratoplasty (Penetrating, lamellar or KPro) is mentioned in the methods, but not described in the results.

Overall, the biggest issue is the use of a poorly defined mixture of techniques (autograft, buccal graft, DPC).  No controls were used, as this was an open case series.

Line 218, change women to woman.  Line 243, change resolved to dissolved. Line 309, change was performed to was not performed.

Minimal English errors.

Reviewer 2 Report

This is an article entitled “Decellularized Porcine Conjunctiva in Treating Severe Symblepharon (jfb-2381550)” which evalautes the efficacy of decellularized porcine conjunctiva 16 (DPC) in treating severe symblepharon.

English needs majör revision. There are many grammatical and typographic error.

Abstract

-          Good.

Introduction

-          Okay.

Materials&Methods

-          Please give the special names etc of the drugs you used.

-          Why did not you initiate corticosteroid drops just after the surgery but after 1 week?

-          Did you perform subconjunctival injections of any kind?

-          Please compare the termal and chemical burns. Were there any differences in the success and results?

-          It would be better to have a control group to compare the differences of the results.

Results

-          What were the chemical burn agents? Please admit.

-          Please add the standart deviations, and please admit what are the numbers you mentioned? Are they means or medians?

Discussion

-          Please discuss the possible side effects of MMC use.

-          Please discuss the important tricks of symbelpharon surgery.

-          Please also discuss the possible porcine tissue mediated complications.

References

-          Good.

Table

-          As you only have 16 cases please make a table just giving data of all cases.

Figures

-          Good.

This is an article entitled “Decellularized Porcine Conjunctiva in Treating Severe Symblepharon (jfb-2381550)” which evalautes the efficacy of decellularized porcine conjunctiva 16 (DPC) in treating severe symblepharon.

English needs majör revision. There are many grammatical and typographic error.

Abstract

-          Good.

Introduction

-          Okay.

Materials&Methods

-          Please give the special names etc of the drugs you used.

-          Why did not you initiate corticosteroid drops just after the surgery but after 1 week?

-          Did you perform subconjunctival injections of any kind?

-          Please compare the termal and chemical burns. Were there any differences in the success and results?

-          It would be better to have a control group to compare the differences of the results.

Results

-          What were the chemical burn agents? Please admit.

-          Please add the standart deviations, and please admit what are the numbers you mentioned? Are they means or medians?

Discussion

-          Please discuss the possible side effects of MMC use.

-          Please discuss the important tricks of symbelpharon surgery.

-          Please also discuss the possible porcine tissue mediated complications.

References

-          Good.

Table

-          As you only have 16 cases please make a table just giving data of all cases.

Figures

-          Good.

Reviewer 3 Report

The authors explore an interesting approach to severe symblepharon treatment. The paper is well written and understandable. Nevertheless, some minor points should be worked on before the manuscript is published.
In general: there is no mention of why xenogeneic material is used. Why did the authors not decellularize human material. This should be briefly addressed/discussed.
Other tasks:
Page 4, Figure 2: please match/explain the colors.
Page 6, line 240 and line 254: is it the same patient? Then this should be mentioned, also presented more clearly in the discussion
Page 9, lines 315 - 320 studies are mentioned, the corresponding references are missing
line 334 AMT should be written out, as it has not been mentioned before.
